# Cancer-Related Neuropathic Pain, Chemotherapy-Induced Peripheral Neuropathy and Cognitive Decline in a 5-Year Prospective Study of Patients with Breast Cancer—NEON-BC

**DOI:** 10.3390/healthcare11243132

**Published:** 2023-12-09

**Authors:** Susana Pereira, Natália Araújo, Filipa Fontes, Luisa Lopes-Conceição, Teresa Dias, Augusto Ferreira, Samantha Morais, Vítor Tedim Cruz, Nuno Lunet

**Affiliations:** 1EPIUnit—Instituto de Saúde Pública, Universidade do Porto, Rua das Taipas, 135, 4050-600 Porto, Portugal; susana.pereira@ipoporto.min-saude.pt (S.P.); filipa.fontes@ispup.up.pt (F.F.); samantha.morais@ispup.up.pt (S.M.); vitor.tedimcruz@ulsm.min-saude.pt (V.T.C.); nlunet@med.up.pt (N.L.); 2Laboratório para a Investigação Integrativa e Translacional em Saúde Populacional (ITR), Rua das Taipas, 135, 4050-600 Porto, Portugal; 3Instituto Português de Oncologia do Porto, Rua Dr. António Bernardino de Almeida, 4200-072 Porto, Portugal; tdjmdias@gmail.com (T.D.); augusto.carmo.ferreira@ipoporto.min-saude.pt (A.F.); 4Departamento de Ciências da Saúde Pública e Forenses e Educação Médica, Faculdade de Medicina da Universidade do Porto, 4200-319 Porto, Portugal; 5Oncology Nursing Research Unit IPO Porto Research Center (CI-IPOP), Portuguese Oncology Institute of Porto (IPO Porto)/Porto Comprehensive Cancer Centre (Porto.CCC) & RISE@CI-IPOP (Health Research Network), 4200-072 Porto, Portugal

**Keywords:** breast neoplasms, neuropathic pain, peripheral nervous system diseases/chemically induced, cognition disorders

## Abstract

This study aims to estimate the prevalence and to identify the determinants of cancer-related neuropathic pain (CRNP), chemotherapy-induced peripheral neuropathy (CIPN) and cognitive decline among patients with breast cancer over five years after diagnosis. Women with an incident breast cancer (n = 462) and proposed for surgery were recruited at the Portuguese Institute of Oncology-Porto in 2012 and underwent systematic neurological examinations and evaluations with the Montreal Cognitive Assessment (MoCA) before treatment and after one, three, and five years. Multivariate logistic regression was used to assess the determinants of CRNP and CIPN, and multivariate linear regression for the variation in MoCA scores. Prevalence of CRNP and CIPN decreased from the first to the fifth year after diagnosis (CRNP: from 21.1% to 16.2%, *p* = 0.018; CIPN: from 22.0% to 16.0% among those undergoing chemotherapy, *p* = 0.007). Cognitive impairment was observed in at least one assessment in 17.7% of the women. Statistically significant associations were observed between: cancer stage III and both CRNP and CIPN; triple negative breast cancer, chemotherapy, axillary node dissection, older age, higher education, and being single and CRNP; taxanes and fruit and vegetable consumption and CIPN. Anxiety, depression and poor sleep quality at baseline were associated with decreases in MoCA values from pre- to post-treatment and with CRNP. Follow-up protocols should consider the persistence of CRNP, CIPN, and cognitive impairment for several years following diagnosis.

## 1. Introduction

In high-income countries, breast cancer is the most prevalent type in women, reflecting its high incidence and survival. Breast cancer ranks first in incidence worldwide [1] and the five-year net survival is over 80% in most of the high-income countries [2]. This justifies the concern about the burden associated with the long-term effects of cancer and its treatment, including neurological complications, among survivors.

Breast cancer-related neurological complications may result from direct nervous system invasion, namely by metastatic disease [3], as well as from indirect nervous system effects, including treatment-related neurological complications. The latter are a growing concern due to their potential to decrease the quality of life or even limit breast cancer treatments among the growing population of cancer survivors [4,5]. Neuropathic pain, chemotherapy-induced peripheral neuropathy (CIPN) and cognitive impairment are potential breast cancer treatment-related complications [6,7,8].

Regarding neuropathic pain due to breast cancer (CRNP), data are available mostly from studies using only screening questionnaires to assess the outcome, and evidence regarding long-terms effects is scarce and mostly from cross-sectional study designs [6,9], which are limited in their ability to identify risk factors for CRNP. Indeed, a systematic review on persistent post-surgical pain following breast cancer surgery reported that only 5% of the studies identified pain based on a physician assessment; although, there are none with a comprehensive neurological exam and a pre-treatment evaluation [9], which are essential for a more robust classification of the type of pain and its relation with cancer and cancer treatment.

Acute and short-term effects have been described for CIPN, namely after a follow-up of up to six months [8], and up to one, two, and three years [9,10,11,12,13,14,15,16]. One study reported a prevalence of CIPN of 47% at an average of six years from cancer diagnosis, but the breast cancer women were those enrolled in four clinical exercise intervention trials (older than 50 years of age, insufficiently active), and therefore, were not representative of six-year breast cancer survivors [12,17].

Likewise, longitudinal studies allowing for the assessment of cognitive decline over several years since breast cancer diagnosis are scarce and present conflicting results. There are reports showing no evidence of an association between chemotherapy [18,19] or hormone therapy [20] and cognitive decline. However, there are studies suggesting a positive association of antineoplastic drugs [21,22] and anastrazole with cognitive deterioration [23].

We have previously presented results from a cohort of women with incident breast cancer [24], and showed that cancer-related neurological complications were frequent, even three years after cancer diagnosis [25]. Breast cancer survivors are frequently discharged from the hospital after five years of follow-up, with the transition of care to primary care units. It is therefore important to quantify the persistence and severity of CRNP and CIPN and the occurrence of cognitive impairment at five years to inform both oncology teams and healthcare professionals at primary care units to adequately plan and manage clinical follow-up during survivorship. As robust information is lacking, both in terms of number of studies and methodology used, or it has limited generalizability, here we update the previous report by quantifying the prevalence and severity of CRNP and CIPN, and the prevalence of cognitive impairment up to five years after diagnosis of breast cancer, as well as by assessing their determinants, based on a systematic neurological exam, including a pre-treatment evaluation. 

## 2. Materials and Methods

### 2.1. Patients and Setting

This study took place at the Portuguese Institute of Oncology—Porto (IPO), the largest cancer hospital in the northern region of Portugal delivering care to patients admitted after a referral from the family doctor.

NEON-BC is a prospective cohort study designed to evaluate the neurological complications of breast cancer; the study protocol has been described elsewhere [26]. Figure 1 represents the assembling of the cohort and the timing of follow-up evaluations. Briefly, between January and December 2012, all women admitted to IPO with a recent diagnosis of breast cancer, proposed for surgery and expected to be followed at IPO were eligible. Those who had been submitted to breast surgery for benign conditions, or to chemotherapy or radiotherapy to the chest for another primary cancer were excluded. Patients with breast cancer who were illiterate or scored less than 17 (or 16 for those aged 65 years or older) in the Portuguese version of the Montreal Cognitive Assessment (MoCA) [27,28,29] were also excluded, considering that they were less likely to be able to answer to questionnaires assessing pain, anxiety and depression.

A total of 506 women were evaluated at baseline, before any treatment for breast cancer, and at one (n = 503), three (n = 475), and five years (n = 466) since breast cancer diagnosis; a total of 464 participants were evaluated in all moments. Reasons for losses to follow-up were: 18 died (the cause of death was neurological in six: meningeal carcinomatosis in two, systemic and cerebral metastases in two, limbic encephalitis and cerebral metastasis in one each), 12 abandoned the study, four transferred to another hospital, two were considered unable to participate by the neurologist and four could not be contacted. The 42 participants lost to follow-up were not significantly different from included participants regarding age (mean 57.4 years vs. 54.5 years, *p* = 0.103), and education (mean 6.9 years vs. 7.7 years, *p* = 0.227), though presented less often with early stage (0, I, II, IIIA) breast cancer (87.8% vs. 95.7%, *p* = 0.026). Additionally, the two participants with stage IV at baseline were not included in data analysis. 

### 2.2. Data Collection

Face-to-face interviews of the participants were conducted by trained interviewers who collected socio-demographic and lifestyles data using a structured questionnaire. Date of birth, number of complete years of education, marital status (single, married or living with partner, divorced, widow), employment (full- and part-time employment, unemployed, retired, sick leave, housewife, student), alcohol consumption, name of regular medication, and previous diagnosis of hypertension and diabetes were self-reported by participants at baseline. As the participants were evaluated between three to six times over the first year of follow-up, and the first year of treatment was full of treatments, exams and consultations, some sociodemographic and lifestyle characteristics were not evaluated at baseline so as not to overwhelm participants with a long evaluation. Indeed, individual monthly income (500 € or lower, 501–1000 €, 1001–1500 €, 1500–2000 €, and >2000 €), weight and height, and lifestyles before breast cancer diagnosis (smoking habits, number of fruit and vegetables pieces consumed per day, and playing a sport) were assessed retrospectively by self-report at the three-year evaluation. 

Participants completed the Hospital Anxiety and Depression Scale (HADS) [30,31] at baseline to measure the levels of anxiety and depression, in the previous week; anxiety and depression sub-scores equal to or higher than 11 out of a possible 21 were considered indicative of clinically significant anxiety or depression, respectively. They also answered the Pittsburgh Sleep Quality Index [32,33,34] to assess sleep quality in the previous month; those with scores equal to or higher than five were classified as having poor quality of sleep.

Clinical data on the tumor and treatments were retrieved from medical records. Classification of cancer stage based on size and extent of the main tumor (T), and spreading of the tumor to lymph nodes (N) and to other parts of the body (metastases, M) followed the TNM classification via the System of the American Joint Committee on Cancer (AJCC), 7th Ed. [35]. Breast cancer subtypes were based on information from medical files regarding immunohistochemistry and in situ hybridization-based biomarkers, namely hormone receptors (HR; estrogen and progesterone receptors, considered positive if present in ≥1% of cells) and human epidermal growth factor receptor (HER2), and were classified into HR-positive/HER2-negative; HER2-positive; and triple-negative breast cancer (HR-negative/HER2-negative).

### 2.3. Assessment of Neurological Complications

Patients were observed by a neurologist who conducted a systematic neurological exam before any cancer treatments (baseline), two weeks after surgery, three weeks after complete chemotherapy (if applicable), and at one, three, and five years after enrollment and additionally, whenever a neurological complication was suspected. The first neurological evaluation was aimed to exclude neurological disease already present before treatments and the after-surgery and after-chemotherapy examinations were intended to assess, respectively, CRNP after surgery, and CIPN and CRNP secondary to chemotherapy. Newly occurring cases of neurological complications were identified through referral by any member of the clinical team, during the systematic neurological evaluations, and additionally, patients were advised to contact the research team directly if symptoms suggestive of neurological pathology appeared. 

During the study period, two neurologists participated in the study, and one was more directly involved in this study and performed 79.6% of the five-year evaluations. 

CIPN was defined as peripheral neuropathy diagnosed after chemotherapy or worsening of a preexisting neuropathy after chemotherapy. The neurologist evaluation included strength, deep tendon reflexes, vibration sensibility (128-Hz tuning fork), and pain sensation (wood cocktail-stick) evaluation. CIPN was classified using the Total Neuropathy Score, clinical version (TNSc) [36] and the Common Terminology Criteria for Adverse Events, V.4.0 (CTCAE) [37].

Neuropathic pain diagnosis was established by the neurologist, after patient examination. Neuropathic pain was classified as probable, according to the Neuropathic Pain Special Interest Group (NeuPSIG) criteria [36], if it had a neuroanatomical plausible distribution, the history suggested a lesion or disease affecting the somatosensory system and the neurologic examination points negative or positive sensory signs in innervation territory of the injured nervous structure. The different locations of pain were registered and pain severity was scored with the Brief Pain Inventory Short Form [38]. This consists of a mean score of four questions measuring the worst, least, average and current pain in the past 24 h (range: 0 to 10, with 0 = “no pain” and 10 = “pain as bad as you can imagine”). In the present study, only CRNP was analyzed. Therefore, only neuropathic pain which onset was after breast cancer treatment and in locations related to the breast and axillary surgeries and to radiotherapy-treated areas was considered, as well as locations related to breast reconstruction surgery, and neuropathic pain due to CIPN (one women with pain in the feet and another one in tips of fingers and toes).

Cognitive performance was assessed with MoCA before cancer treatment, and after one, three, and five years. Cognitive impairment was considered present if the participant scored at least two standard deviations below the mean of age- and education-specific distributions from normative data [27].

### 2.4. Statistical Analysis

Characteristics of the patients and their lifestyle, of the tumor, and of the treatments received were presented as counts and proportions.

For each neurological complication, point prevalences at the follow-up evaluations and period prevalences over the five years were computed; for cognitive impairment, the percentages of participants with cognitive impairment at baseline and at each of the follow-up assessments were estimated. Comparisons between different moments of evaluation were performed using the McNemar’s test.

Considering the two dichotomic variables CRNP present/absent and CIPN present/absent at each of the three follow-up evaluations, there are eight possible individual trajectories for the presence of CRNP and CIPN over time, which were grouped as follows: (1) presenting the neurological complication at least once over time; (2) CRNP/CIPN present at the five-year evaluation; and (3) constant presence of the neurological complication at each evaluation. For CRNP and CIPN, each of these three groups were compared with the reference group, which included the participants who never had the neurological complication. Adjusted odds ratios (OR) were computed using logistic regression, to quantify the association between participants’ characteristics, clinical data of the tumor and treatments received, with the presence of CRNP and CIPN over the five-year follow-up. A multiple linear regression analysis was used to estimate β coefficients of the relation between participants’ characteristics, clinical data of the tumor, and treatments performed, with the variation in the MoCA score between baseline assessment and each follow-up evaluation. This considered only participants who did not present cognitive impairment at baseline. Variables introduced in the multivariate logistic regression or linear regression models to compute the adjusted odds ratio and the adjusted β coefficients, respectively, are described in the footnotes of the tables. Assumptions required to use logistic and linear regression models were verified. Values of the variance inflation factor were up to 2.02. The linktest command was used to assess if the models were correctly specified.

Statistical analyses were performed using Stata, version 15.1 (StataCorp, College Station, TX, USA) and a significance level of 0.05 was considered.

### 2.5. Ethics

The study was approved by the Ethics Committee of the Portuguese Institute of Oncology of Porto (Ref. CES 406/011, CES 99/014, CES 198/016). All participants provided written informed consent.

## 3. Results

At baseline, nearly half of the women were 55 years or older, 42.0% had up to four years of education and 29.2% had more than 10 years. Cancer stage was 0 or I for 55.0% of the women, while 30.3% and 14.7% of the patients presented with stages II and III, respectively. More than three quarters of the participants had HR+/HER2- breast cancer subtype, 14.8%, HER2+, and 8.3%, triple negative. Most women (94.6%) received cancer treatment only during the first year following diagnosis. Just over half of the women underwent breast-conserving surgery. Chemotherapy was used in 60.3% of the patients, radiotherapy in 73.8%, hormone therapy in 84.0% and targeted therapy in 13.2% (Table 1).

A total of 29 patients received additional treatment between the first and the fifth year of follow-up, due to recurrence of breast cancer (n = 11) or an incident second primary cancer (n = 19) (Appendix A). 

At the baseline evaluation, 65 women had low cognitive performance, 41 had migraines, 19, tremor, eight, tension headaches, three, diabetic polyneuropathy, three, carpal tunnel syndrome, one, tomaculous neuropathy, and other neurologic conditions were also observed in one to three women (Appendix A).

### 3.1. Neuropathic Pain

The prevalence of CRNP was 21.0% (95% CI: 17.4%,25.0%) at the one-year evaluation, 24.0% (95% CI: 20.2%,28.2%) after three years, and 16.2% (95% CI: 13.0%,19.9%) after five years (Figure 2). A total of 35.1% (95% CI: 30.7%,39.6%) of the participants presented CRNP at least once over the five years of follow-up, 7.8% in all evaluations and 16.7% in only one (Appendix A).

Among those presenting CRNP at the three evaluations, the median pain severity score increased significantly from the one- to the three-year evaluation (2.5 vs. 3.6, *p* = 0.006) and there was no significant change between the third and the fifth year after breast cancer diagnosis (3.6 vs. 3.5, *p* = 0.640). Similarly, the mean of the maximum pain felt in the past 24 h increased significantly from the one- to the three-year evaluation (4.6 vs. 6.2, *p* < 0.001) and no significant change was observed between the third and the fifth year after breast cancer diagnosis (6.2 vs. 6.3, *p* = 0.836). Women who presented CRNP only once had a median pain severity score lower than women with CRNP more than once (1.0 vs. 2.4, *p* < 0.001 at one year, 2.5 vs. 3.3, *p* = 0.002 at three years, and 2.4 vs. 3.5, *p* = 0.050, at five years). Similarly, the mean of the maximum pain felt in the last 24 h was lower for women with CRNP present only once over time (3.4 vs. 4.3, *p* = 0.021 at one-year, 4.8 vs. 6.0, *p* = 0.006, at three-years and 4.6 vs. 6.2, *p* = 0.012, at five-years). 

At the five-year evaluation, CRNP was localized in the breast (68% of women with CRNP), the arm (21.3%), the axillary area (16.0%), the chest wall (6.7%), and the inguinal region (2.7%). CRNP presented with pins and needles sensation and hypoesthesia to prick in all women with CRNP, and with hypoesthesia to touch (97.3%), electric shocks sensation (80.0%), itching sensation (73.3%), tingling sensation (61.3%), painful cold sensation (48.0%), burning sensation (37.3%) and allodynia (29.3%).

### 3.2. Chemotherapy-Induced Peripheral Neuropathy

Among women who underwent chemotherapy over the five years, the prevalence of CIPN was 22.1% (95% CI: 17.4%,27.4%), 19.2% (95% CI: 14.8%,24.3%), and 16.0% (95% CI: 11.9%,20.8%) after one, three, and five years of follow up, respectively (Figure 2). A total of 26.3% of participants had CIPN at least once during the follow-up period, 11.7% in all evaluations and 7.1% in only one evaluation (Appendix A).

Among those presenting CIPN in the three evaluations, the median TNSc scores decreased non-significantly from the one- to the three-year evaluation (5.0 vs. 4.0, *p* = 0.075) and decreased significantly between the third and the fifth year after breast cancer diagnosis (4.0 vs. 3.0, *p* < 0.001). Women who presented CIPN only once had a lower median TNSc score than women with CIPN more than once, (1.5 vs. 5.0 at the one-year evaluation, *p* < 0.001), but no significant differences were observed at the three and five-year evaluations (10.0 [n = 1] vs. 4.0, *p* = 0.109, after three years, and 4.0 vs. 4.0, *p* = 0.089, after five years). Among women presenting CIPN in the three evaluations, peripheral sensory neuropathy grades one or two of the CTCAE classification were observed for 100%, 97.0%, and 97.0% of the cases after one, three, and five years of follow-up, respectively. Peripheral motor neuropathy was less frequent, with grades one or two present in 9.1%, 15.2%, and 15.2% of the women, at the one-, three- and five-year evaluations, respectively.

Women with CIPN and CRNP (n = 15) accounted for 33.3% of the participants with CIPN.

### 3.3. Cognitive Performance Assessed Using MoCA

Cognitive impairment affected 7.8% (95% CI: 5.5%, 10.6%) of women with breast cancer before any treatment and its prevalence remained stable over the five years: 6.7% (95% CI: 4.6%, 9.4%), 7.8% (95% CI: 5.5%, 10.6%), and 7.6% (95% CI: 5.3%, 10.4%) at years one, three, and five, respectively (Figure 2). The proportion of women with very low MoCA score (<16 or <17 for women aged 65 or older) was 0% at baseline, as these women were excluded from the cohort, and 1.7% (95% CI: 0.07%, 3.4%), 4.5% (95% CI: 2.8%, 6.8%), and 4.3% (95% CI: 2.7%, 6.6%), at one-, three-, and five-years. A total of 17.7% (95% CI: 14.4%, 21.5%) of the women presented cognitive impairment at least once during the five years.

From the 36 women with cognitive impairment at baseline, 14 recovered at the one-year evaluation and performed within the normal range of the MoCA over time, four presented with a low MoCA score at all evaluations, and 18 had cognitive impairment in at least one of the follow-up assessments. Among women with no cognitive impairment at baseline, 89.2% presented a MoCA score within the normal range throughout the whole follow-up, and the remaining had cognitive impairment at least once (Appendix A).

The mean MoCA scores increased from 23.3 at baseline to 24.0 at one year (*p* < 0.001), followed by a decrease to 23.6 at three years (*p* < 0.001) and was 23.7 at the end of follow-up (*p* = 0.144).

### 3.4. Factors Associated with CRNP, CIPN and Variation in Cognitive Performance

In Appendix A present sociodemographic characteristics, lifestyles, and co-morbidities of participants, according to the presence of CRNP (Appendix A) and CIPN (Appendix A). Statistically significant differences were further studied with multivariable logistic regression models.

Cancer stage III, triple negative breast cancer and chemotherapy were significantly associated with CRNP, either present at least once during the five years of follow-up, at five years, or in all evaluations (adjusted OR [aOR] ranged from 2.02 to 4.04). Anxiety, depression and poor sleep quality were also positively associated with CRNP (aOR between 2.24 and 6.13). Associations between patients aged 55 or older (OR= 0.61, 95% CI: 0.42,0.90), those with at least ten years of education (aOR= 0.59, 95% CI: 0.35,0.99), those who were single (aOR = 0.45, 95% CI: 0.22, 0.94), and those who had axillary node dissection (aOR = 2.11, 95% CI: 1.13, 3.03) with CRNP was significant only for the group of participants with CRNP present at least once over the follow-up period (Table 2).

For chemotherapy-induced peripheral neuropathy, only participants who underwent chemotherapy were considered (N = 281). Cancer stage III and treatment with taxanes were associated with CIPN, whether this complication was present only once during the five years of follow-up, or at five years, or at all assessments (aOR ranging between 3.63 and 12.69; Table 3). The consumption of at least five portions per day of fruits and vegetables was negatively associated with CIPN present at all evaluations (aOR = 0.25, 95% CI: 0.07, 0.87).

Table 4 describes MoCA changes from baseline to year one, three, and five in participants without probable cognitive impairment at baseline. Being 65 years or older was negatively associated with variations in the MoCA score between the baseline and the one-year evaluation, and the baseline and the five-year assessment (β = −0.74 and β = −0.87, respectively, *p* < 0.050), while higher education was positively associated with changes in cognitive performance from baseline to the follow-up evaluations (adjusted β coefficients ranging from 0.91 to 2.38, *p* < 0.010). Significant negative associations were observed between anxiety, depression and poor sleep quality with the variation in MoCA score (adjusted β coefficients between −1.60 and −0.63).

## 4. Discussion

Our results show that neurological complications are frequent in the first five years after breast cancer diagnosis, and long-lasting effects of CRNP and CIPN were observed over the five years. Nearly one in every five participants had cognitive impairment at least once during the follow-up. Clinical characteristics of the breast cancer and its treatment were associated with CIPN and CRNP, but not with cognitive decline. While patients’ characteristics at baseline, namely, anxiety, depression and poor sleep quality, were associated with CRNP and cognitive decline, but not with CIPN, except the consumption of fruit and vegetables for persistent CIPN.

CRNP was the most frequent treatment-related neurological complication throughout the follow-up. Despite the median pain severity scores at the fifth year being only 3.5, the mean of the maximum pain felt in the previous 24 h was 6.3, reflecting the paroxysmal character of neuropathic pain. A recent systematic review and meta-analysis reported the prevalence of neuropathic pain after breast cancer treatment [6]. Among the studies identified, two [39,40] had a follow-up time similar to our study but in one [40], only women submitted to axillary lymph node dissection were included and this surgery is associated with higher odds of CRNP; in the other study, the estimated prevalence was 9.0% [39], but CRNP was assessed with questionnaires, which may explain the lower prevalence compared to our results. Another systematic review and meta-analysis on post-surgical pain in breast cancer reported the prevalence and severity of persistent pain were stable for over two years [9], which is different from our results as we observed an increase in the prevalence and severity at the three-year evaluation and then a stabilization.

We found associations between younger age, axillary node dissection, cancer stage III, triple negative breast cancer, and chemotherapy, with CRNP present at least once over the follow-up period. The same predictors have been described in studies that analyzed pain in general, that did not distinguish between CRNP or nociceptive pain [41,42]. Our results showed that single women were less likely to have CRNP over the five years of follow-up, which was also observed previously [43] but in only one of five studies included in a meta-analysis that concludes with no significant effect of marital status on the occurrence of pain in breast cancer patients [42]. Therefore, the association of marital status with CRNP should be further studied in other studies.

In line with previous studies, taxane-based chemotherapy was strongly associated with CIPN [4,44], but alcohol consumption and diabetes at baseline were not. The latter could be related to limited statistical power due to the low levels of alcohol intake, as well as the fact that diabetes was only controlled with oral medicines in most of the patients (93.5%). However, a positive association between diabetes and CIPN due to cancer has been previously described in colorectal cancer patients [45,46,47]. Participants who reported a consumption of at least five portions of fruits and vegetables were less likely to present CIPN in all assessments. In colorectal patients, consumption of fruits and vegetables pre-treatment were associated with better indicators of health-related quality of life but not with CIPN 24 months after treatment [48], although CIPN was assessed using a questionnaire (EORTC QLQ-CIPN20 of the European Organisation for Research and Treatment of Cancer). In breast cancer patients, no association was also reported in taxane-treated patients after 24 months, although CIPN was assessed using the Functional Assessment of Cancer Therapy–Taxane Neurotoxicity [16], and in taxane- or platinum-treated patients after six months [49]. Our results were only significant for the groups of participants who always presented CIPN over the five years of follow-up which may indicate an association with severe CIPN. This result should be further confirmed in other studies as this could contribute for the design of interventions aiming to reduce CIPN. 

The prevalence of cognitive impairment ranged between 6.9% and 7.8% over the five years, but this disorder affected 17.7% of participants at least once during the five-year follow-up period. Indeed, for most women, cognitive impairment was not consistently observed in all evaluations (Appendix A). This may have several possible explanations; (1) in repeated evaluations, practice effects may mask cognitive decline [50], and a score that remains stable or improves may not correspond to a real improvement in cognitive function; (2) different treatments for breast cancer may affect cognitive performance in different moments, namely, an acute effect at the end of chemotherapy, with a recovery after six months has been reported [51]. This includes a short-term effect of radiotherapy over seven months following treatment and a recovery after three years [52], and short- and long-term effects of hormone therapy [53]; (3) factors, such as anxiety and depression, associated with the experience of a cancer diagnosis and treatment may also have an impact on cognitive assessment, being present to a different extent in different moments of the follow-up; (4) some of these cases of cognitive impairment may also be completely independent of cancer, namely in older women. Our results on the prevalence of cognitive impairment at each evaluation are lower than previously reported, namely 28.0% of women with breast cancer before surgery or any other treatment [54], 35.0% of women before adjuvant treatment for breast cancer [55], 16.0% of patients six months after chemotherapy [56], and 19.0% of patients treated with chemotherapy or not, after a median of 17 months since diagnosis [57]. Methodological differences may account for the heterogeneous results; in a previous study [57], the prevalence of cognitive impairment varied between 19.0% and 35.5% depending on the criterion used to define the outcome. Global scores of cognitive performance, such as the MoCA score, are less sensitive to cognitive impairment affecting specific domains. The International Cognition Cancer Task Force recommends the assessment of verbal learning and memory, information processing speed and executive functions as they are cognitive domains that could be most affected by chemotherapy [58]. However, the cognitive domains affected among patients with breast cancer in general and which tests should be used to assess them remain unclear.

Another possible explanation is that 80 patients were excluded from the NEON-BC study, because the MoCA test score suggested cognitive impairment before starting breast cancer treatments. We used this criterion to ensure the reliability of data provided by patients in self-rating scales (such as HADS or the Brief Pain Inventory Short Form) and to exclude primary dementia, not related to cancer; however, this might be a population particularly susceptible to cognitive decline. As other previous studies have described cognitive impairment before treatment, we may have excluded some cases of cognitive impairment in the context of paraneoplastic neurological syndrome. If these 80 women had been included, the prevalence of cognitive impairment at baseline would have been 21.0%. Although we did not evaluate post-traumatic stress disorder, it was also reported as a possible cause of cognitive impairment before any treatment for breast cancer [59]. Anxiety and depression were measured at each evaluation in the NEON-BC cohort using the HADS, which screening accuracy was shown to be high in women with breast cancer [60]. At baseline, there were no statistical differences in terms of anxiety and depression between women with or without cognitive impairment, although we could not evaluate if these differences existed between women with very low MoCA score who were excluded from the NEON-BC cohort with those who participated.

To assess the baseline factors and treatments associated with cognitive performance over time in women with no cognitive impairment at baseline, we used the variation in the MoCA score from baseline to subsequent assessments as the cognitive outcome. This being under the assumption that even when not translating into incident cognitive impairment, less favorable changes in performance may be associated with progressive cognitive deterioration. We identified a negative association between anxiety, depression and poor quality of sleep at baseline and changes in the MoCA score, namely from baseline to the five-year evaluation, which is in accordance with previous studies on cognitive decline conducted in the general population [61,62,63]. We did not identify previous studies analyzing the possible associations of anxiety, depression or sleep quality at baseline with long-term cognitive decline in patients with breast cancer. Our results show that anxiety, depression and poor sleep quality before treatments may be considered important factors to identify groups of women more likely to develop a less favorable change in cognitive performance, even up to five years later. 

The sample size of the NEON-BC study was calculated based on the expected occurrence of CRNP and CIPN and potential associations of factors with a prevalence of at least 10% [26]. Diabetes was the variable with lower prevalence at baseline, 10.0%, which may explain the absence of an association of diabetes with CRNP in this study. Cognitive impairment was less frequent and chemotherapy treatment was performed in more women than expected (60.2% instead of 50%); therefore, this study may not have power to detect lower effect size associations of factors with cognitive outcomes. Physical activity could not be assessed at baseline, rather, the practice of a sport before breast cancer diagnosis was retrospectively reported at the three-year assessment. This may have contributed for the absence of an association between physical activity and CIPN, although such an association has been observed in other studies [64].

To the best of our knowledge, this is the first study providing a prospective and comprehensive assessment of long-term neurological effects of breast cancer management, including CRNP, CIPN and cognitive impairment over five years following a breast cancer diagnosis. The occurrence of these neurological complications was based on a clinical examination by a neurologist, not only self-report of symptoms by patients, and standardized instruments were used to assess cognitive function, CIPN and CRNP. The neurologist was not blinded regarding treatments and clinical characteristics of the patient at the moment of the neurological evaluation. Unfortunately, the team of neurologists was too small to allow a separation between the systematic clinical evaluation within the study and the provision of clinical care to the participant/patient, but standardized procedures were followed to minimize bias. Moreover, the baseline evaluation before cancer treatments, allowed us to exclude neurological conditions not related to cancer. Despite the single-center study design, IPO-Porto is the largest breast cancer oncological center in Portugal, receiving patients from any part of the country. Finally, we only included patients proposed for breast surgery, which limits the generalizability of our results to women with early-stage breast cancer.

For women with a breast cancer history, these results highlight that special care should be sought after discharge of the hospital to get adequate treatment for pain symptoms because it may be CRNP, which needs a different pharmacological treatment. Clinical practitioners should identify patients with anxiety, depression, and poor sleep quality before cancer treatment commences. Adequate management of these conditions should be provided through medication and other interventions due to the suffering they cause and because they are risk factors for long-term CRNP and cognitive decline; younger patients (below 55 years), those presenting cancer stage III, and those with triple negative breast cancer were also more likely to present CRNP, including at five years; probably due to more aggressive treatments. Although higher education and being single were protective factors when considering the presence of CRNP in at least one evaluation, no significant results were observed for CRNP at five years; this may be explained by a lack of power to detect such an association as the proportions of women with more than ten years of schooling and those who were single at baseline were low. Nevertheless, these factors are not likely to be modifiable, but further research should disentangle which mechanisms education and marital status affect CRNP occurrence. Cancer stage III and taxane-based treatment are the main factors influencing CIPN frequency at five years, and perhaps, a low consumption of fruits and vegetables before cancer diagnosis. No other risk factors were identified for CIPN but women treated with chemotherapy were only part of the cohort (279) and our analyses may have been underpowered to identify weaker associations or of less frequent exposures. 

Further research should be conducted to better understand the characteristics and the etiology of cognitive impairment in women with breast cancer, namely, regarding vascular risk factors and neurodegenerative processes. 

## 5. Conclusions

CRNP and CIPN are frequent adverse effects of breast cancer treatments, and they are often long-lasting. Cognitive impairment was often present before treatments and affected nearly 18% of the women over the five years. These results suggest that follow-up protocols should take into account the persistence of these conditions for several years following diagnosis. Special attention is recommended for women presenting cancer stage III and those with triple negative breast cancer, those treated with chemotherapy, and particularly with taxanes. Also, anxiety, depression and poor sleep quality before treatment should be valued as they are associated with both CRNP and less favorable cognitive changes after treatments.

## Figures and Tables

**Figure 1 healthcare-11-03132-f001:**
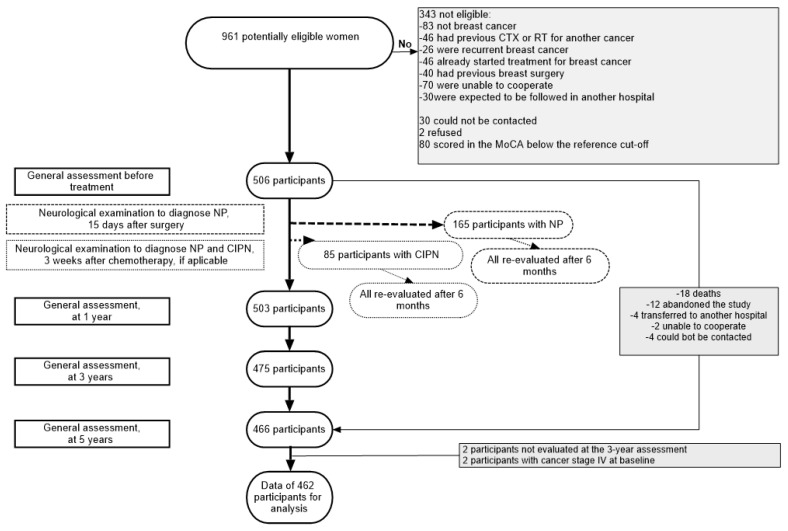
Flow chart describing the assembling of the cohort and the evaluations performed during the follow-up. CIPN, chemotherapy-induced peripheral neuropathy; CTX, chemotherapy; NP, neuropathic pain; RT, radiotherapy.

**Figure 2 healthcare-11-03132-f002:**
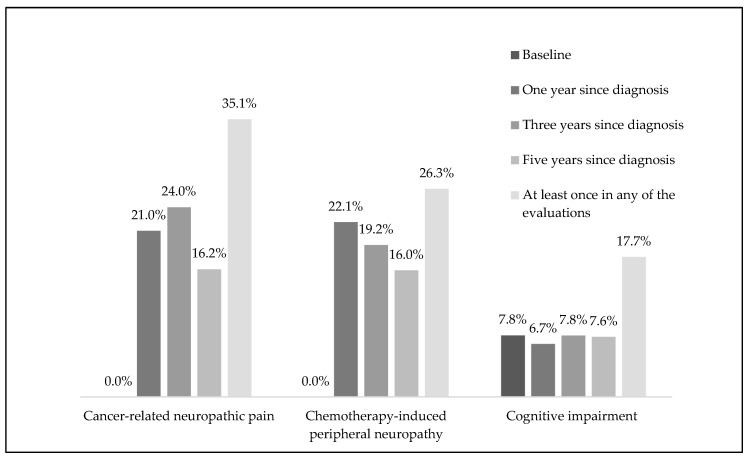
Prevalence of cancer-related neuropathic pain, chemotherapy-induced peripheral neuropathy and cognitive impairment, during the five-years of follow-up.

**Table 1 healthcare-11-03132-t001:** Socio-demographics characteristics of the participants (N = 462), lifestyles, co-morbidities, and clinical characteristics of the oncologic disease, at the baseline evaluation, before any treatment, and treatments performed during the first year after breast cancer diagnosis.

Participants’ and Cancer Characteristics and Treatments		n	%
Socio-demographics			
Age			
	≥55 years	230	49.8
Education (years)			
	Primary education (≤4)	194	42.0
	Lower secondary education (5–9)	133	28.8
	At least upper secondary education (≥10)	135	29.2
	Living in Greater Porto Area	207	44.8
Marital status			
	Married/living together	323	69.9
	Single	49	10.6
	Widow/divorced	90	19.5
Professionally active (n = 460)			
	Yes	242	52.6
Monthly income (n = 454)			
	Above EUR 500 ^a^	204	44.9
Lifestyles			
	Alcohol consumption > 10 g/day (n = 461)	92	20.0
	Past or current smoker	96	20.8
	Fruits and vegetables ≥ 5/day (n = 459)	101	22.0
	Playing a sport	80	17.3
Comorbidities			
	Hypertension	146	31.6
	Diabetes	46	10.0
Body Mass Index (kg/m^2^)			
	<18.5	5	1.1
	18.5–24.5	198	43.0
	25.0–29.9	155	33.6
	≥30	103	22.3
Chronic medicines consumption			
	None	165	35.7
	One	77	16.7
	Two to four	149	32.3
	More than four	71	15.4
Cancer stage			
	0	31	6.71
	I	223	48.27
	II	140	30.3
	III	68	14.7
Breast cancer subtype (n = 433)			
	HR+/HER2-	333	76.9
	HER2+	64	14.8
	Triple negative	36	8.3
Treatments in the first year			
Breast surgery			
	Breast conserving	235	50.9
	Mastectomy	227	49.1
Axillary surgery			
	Sentinel lymph node biopsy	295	65.9
	Axillary lymph node dissection	153	34.1
Chemotherapy (n = 279)			
	Neoadjuvant	30	10.8
	Adjuvant	249	89.2
	Taxane-based	198	71.0
Radiotherapy			
	Yes	341	73.8
Endocrine therapy			
	Yes	388	84.0
Targeted therapy			
	Yes	61	13.2

^a^ EUR500 is the median value of monthly income in the sample. HR+/HER2 stands for tumor expressing hormone receptor but not overexpressing human epidermal growth factor receptor 2; HER2+ stands for tumor overexpressing HER2; triple negative stands for tumor not expressing estrogen receptors, nor progesterone receptors, nor overexpressing HER2.

**Table 2 healthcare-11-03132-t002:** Association between socio-demographic and clinical characteristics of the patients and neuropathic pain (CRNP).

		Reference Group Patients Who Never Had CRNP (N = 301)	Patients with CRNP at Least Once (N = 163)	Those with CRNP at Least Once vs. Reference Group	Patients with CRNP at Five-Years (N = 75)	Those with CRNP at Five-Years vs. Reference Group	Patients with CRNP in All Evaluations (N = 36)	Those with CRNP in All Evaluations vs. Reference Group
		n (%)	n (%)	Adjusted OR [95% CI]	n (%)	Adjusted OR [95% CI]	n (%)	Adjusted OR [95% CI]
Age (years)								
	<55	138 (59.5)	94 (40.5)	ref.	44 (19.0)	ref.	18 (7.8)	ref.
	≥55	162 (70.4)	68 (29.6)	0.62 * [0.42, 0.91]	31 (13.5)	0.60 [0.36, 1.00]	18 (7.8)	0.85 [0.43, 1.70]
Education (years)								
	≤4	122 (62.9)	72 (37.1)	ref.	33 (17.0)	ref.	19 (9.8)	ref.
	5–9	87 (65.4)	46 (34.6)	0.66 [0.40, 1.10] ^e^	24 (18.0)	0.77 [0.40, 1.46] ^e^	9 (6.8)	0.56 [0.23, 1.37] ^e^
	≥10	91 (67.4)	44 (32.6)	0.59 * [0.35, 0.99] ^e^	18 (13.3)	0.54 [0.27, 1.08] ^e^	8 (5.9)	0.47 [0.19, 1.19] ^e^
Marital status								
Married/living together		200 (66.7)	123 (75.9)	ref.	52 (69.3)	ref.	25 (69.4)	ref.
Single		39 (13.0)	10 (6.2)	0.45 * [0.22, 0.94] ^f^	8 (10.7)	0.87 [0.38, 2.00] ^f^	4 (11.1)	0.90 [0.29, 2.77] ^f^
Widower/divorced		61 (20.3)	29 (17.9)	0.84 [0.51, 1.40] ^f^	15 (20.0)	1.05 [0.54, 1.78] ^f^	7 (19.4)	0.92 [0.37, 2.29] ^f^
Cancer stage								
	0/I	179 (70.5)	75 (29.5)	ref.	32 (12.6)	ref.	16 (6.3)	ref.
	II	91 (65.0)	49 (35.0)	1.29 [0.82, 2.02] ^f^	26 (18.6)	1.56 [0.87, 2.81] ^f^	10 (7.1)	1.29 [0.55, 2.99] ^f^
	III	30 (44.1)	38 (55.9)	2.95 *** [1.68, 5.18] ^f^	17 (25.0)	3.04 ** [1.48, 6.20] ^f^	10 (14.7)	3.82 ** [1.56, 9.33] ^f^
Breast cancer subtypes ^a^								
	HR+/HER2	219 (65.8)	114 (34.2)	ref.	51 (15.3)	ref.	22 (6.6)	ref.
	HER2+	45 (70.3)	19 (29.7)	0.82 [0.45, 1.47] ^f^	7 (10.9)	0.68 [0.29, 1.61] ^f^	4 (6.3)	0.95 [0.31, 2.93] ^f^
	Triple negative	17 (47.2)	19 (52.8)	2.02 * [1.00, 4.07] ^f^	11 (30.6)	2.60 * [1.14, 5.95] ^f^	7 (19.4)	4.04 ** [1.49, 10.97] ^f^
Breast surgery								
	Breast-conserving	161 (68.5)	74 (31.5)	ref.	34 (14.5)	ref.	17 (7.2)	ref.
	Mastectomy	139 (61.2)	88 (38.8)	1.19 [0.77, 1.83] ^g^	41 (18.1)	1.13 [0.63, 2.02] ^g^	19 (8.4)	0.73 [0.32, 1.67] ^g^
Axillary surgery ^b^								
	SLNB	212 (71.9)	83 (28.1)	ref.	39 (13.2)	ref.	18 (6.1)	ref.
	ALND	79 (51.6)	74 (48.4)	2.11 * [1.13, 3.93] ^g^	34 (22.2)	1.80 [0.81, 4.03] ^g^	18 (11.8)	2.67 [0.80, 8.93] ^g^
Chemotherapy								
	No	135 (74.6)	46 (25.4)	ref.	19 (10.5)	ref.	9 (5.0)	ref.
	Yes	165 (58.7)	116 (41.3)	2.05 * [1.19, 3.53] ^g^	56 (19.9)	2.69 * [1.24, 5.83] ^g^	27 (9.6)	3.40 * [1.16, 9.93] ^g^
Radiotherapy								
	No	84 (70.6)	35 (29.4)	ref.	19 (16.0)	ref.	8 (6.7)	ref.
	Yes	216 (63.0)	127 (37.0)	1.16 [0.55, 2.42] ^h^	56 (16.3)	0.67 [0.26, 1.77] ^h^	28 (8.2)	0.61 [0.15, 2.58] ^h^
Anxiety ^c^								
	No	208 (73.5)	75 (26.5)	ref.	28 (9.9)	ref.	11 (3.9)	ref.
	Yes	91 (51.1)	87 (48.9)	2.72 *** [1.80, 4.12] ^i^	47 (26.4)	3.95 *** [2.26, 6.90] ^i^	25 (14.0)	6.02 *** [2.66, 13.6] ^i^
Depression ^c^								
	No	287 (67.7)	137 (32.3)	ref.	59 (13.9)	ref.	27 (6.4)	ref.
	Yes	13 (34.2)	25 (65.8)	3.91 *** [1.90, 8.02] ^i^	16 (42.1)	6.13 *** [2.67, 14.12] ^i^	9 (23.7)	10.95 *** [3.79, 31.65] ^i^
Poor sleep quality ^d^								
	No	112 (75.7)	36 (24.3)	ref.	11 (7.4)	ref.	5 (3.4)	ref.
	Yes	187 (59.7)	126 (40.3)	2.24 ** [1.42, 3.54] ^i^	64 (20.4)	4.13 *** [2.03, 8.39] ^i^	31 (9.9)	4.19 ** [1.54, 11.44] ^i^

ALND, axillary lymph node dissection; CI, confidence interval; CRNP, cancer-related neuropathic pain; OR, odds ratio; SLNB, sentinel lymph node biopsy. * *p* < 0.05, ** *p* < 0.01, *** *p* < 0.001. ^a^ This information is missing for 29 participants. ^b^ Patients who had both ALND and SLNB are reported as ALND; N < 462, because 14 patients only performed breast surgery. ^c^ Baseline depression and anxiety were defined as presenting the respective sub-score equal to or higher than 11 in the Hospital Anxiety and Depression Scale. ^d^ Poor quality of sleep at baseline was defined as presenting a total score equal to or higher than five in the Pittsburg Sleep Quality Index. ^e^ Adjusted for age. ^f^ Adjusted for age and education. ^g^ Adjusted for age, education, cancer stage and breast cancer subtypes. ^h^ Adjusted for age, education, cancer stage, breast cancer subtypes, breast and axillary surgeries. ^i^ Adjusted for age, education and cancer stage.

**Table 3 healthcare-11-03132-t003:** Association between socio-demographic, lifestyle, clinical and treatment characteristics of the patients among those who were submitted to chemotherapy, and chemotherapy-induced peripheral neuropathy (CIPN).

		Reference Group Patients Who Never Had CIPN (N = 207)	Patients with CIPN at Least Once (N = 74)	Those with CIPN at Least Once vs. Reference Group	Patients with CIPN at Five-Years (N = 45)	Those with CIPN at Five-Years vs. Reference Group	Patients with CIPN at All Evaluations (N = 33)	Those with CIPN at All Evaluations vs. Reference Group
		n (%)	n (%)	Adjusted OR [95% CI]	n (%)	Adjusted OR [95% CI]	n (%)	Adjusted OR [95% CI]
Age (years)								
	≤55	123 (74.5)	42 (25.5)	ref.	22 (13.3)	ref.	15 (9.1)	ref.
	>55	84 (72.4)	32 (27.6)	1.12 [0.65, 1.91]	23 (19.8)	1.53 [0.80, 2.92]	18 (15.5)	1.76 [0.84, 3.68]
Education (years)								
	≤4	72 (73.5)	26 (26.5)	ref.	14 (14.3)	ref.	10 (10.2)	ref.
	5–9	77 (77.0)	23 (23.0)	0.87 [0.44, 1.73] ^c^	15 (15.0)	1.26 [0.54, 2.96] ^c^	11 (11.0)	1.39 [0.52, 3.67] ^c^
	≥10	58 (69.9)	25 (30.1)	1.25 [0.63, 2.50] ^c^	16 (19.3)	1.75 [0.75, 4.08] ^c^	12 (14.5)	1.95 [0.75, 5.10] ^c^
Diabetes at baseline								
	No	190 (72.8)	71 (27.2)	ref.	43 (16.5)	ref.	31 (11.9)	ref.
	Yes	17 (85.0)	3 (15.0)	0.41 [0.11, 1.49] ^d^	2 (10.0)	0.43 [0.09, 2.03] ^d^	2 (10.0)	0.59 [0.12, 2.83] ^d^
Alcohol consumption at baseline								
	<10 g/day	166 (73.5)	60 (26.5)	ref.	35 (15.5)	ref.	24 (10.6)	ref.
	≥10 g/day	41 (74.5)	14 (25.5)	0.99 [0.49, 1.97] ^d^	10 (18.2)	1.26 [0.56, 2.84] ^d^	9 (16.4)	1.71 [0.71, 4.12] ^d^
Daily consumption of fruits and vegetables								
	Less than 5 portions	155 (74.9)	61 (82.4)	ref.	38 (84.4)	ref.	30 (90.9)	ref.
	At least 5 portions	52 (25.1)	13 (17.6)	0.61 [0.31, 1.20] ^d^	7 (15.6)	0.48 [0.20, 1.17] ^d^	3 (9.1)	0.25 * [0.07, 0.87] ^d^
Cancer stage								
	0/I	76 (80.9)	18 (19.1)	ref.	9 (9.6)	ref.	4 (4.3)	ref.
	II	95 (77.9)	27 (22.1)	1.24 [0.63, 2.44] ^d^	17 (13.9)	1.63 [0.68, 3.92] ^d^	15 (12.3)	3.32 * [1.04, 10.61] ^d^
	III	36 (55.4)	29 (44.6)	3.63 *** [1.76, 7.47] ^d^	19 (29.2)	5.07 *** [2.04, 12.63] ^d^	14 (21.5)	8.75 *** [2.60, 29.41] ^d^
Breast cancer subtypes								
	HR+/HER2	145 (77.5)	42 (22.5)	ref.	25 (13.4)	ref.	18 (9.6)	ref.
	HER2+	41 (65.1)	22 (34.9)	1.84 [0.99, 3.44] ^d^	12 (19.0)	1.63 [0.75, 3.56] ^d^	11 (17.5)	2.10 [0.91, 4.86] ^d^
	Triple negative	21 (67.7)	10 (32.3)	1.72 [0.75, 3.96] ^d^	8 (25.8)	2.46 [0.97, 6.27] ^d^	4 (12.9)	1.70 [0.52, 5.61] ^d^
Taxanes-based chemotherapy								
	No taxanes	74 (96.1)	3 (3.9)	ref.	2 (2.6)	ref.	1 (1.3)	ref.
	Taxanes	133 (65.2)	71 (34.8)	12.69 *** [3.45, 46.74] ^e^	43 (21.1)	8.79 ** [1.80, 42.97] ^e^	32 (15.7)	8.77 * [1.04, 73.60] ^e^
5-FU-based chemotherapy								
	No 5-FU	71 (78.0)	20 (22.0)	ref.	14 (15.4)	ref.	8 (8.8)	ref.
	5-FU	136 (71.6)	54 (28.4)	1.45 [0.75, 2.80] ^e^	31 (16.3)	1.05 [0.48, 2.30] ^e^	25 (13.2)	1.31 [0.51, 3.38] ^e^
Anxiety ^a^								
	No	128 (74.9)	43 (25.1)	ref.	31 (18.1)	ref.	22 (12.9)	ref.
	Yes	78 (71.6)	31 (28.4)	1.28 [0.73, 2.25] ^f^	14 (12.8)	0.79 [0.38, 1.62] ^f^	11 (10.1)	0.84 [0.37, 1.89] ^f^
Depression ^a^								
	No	193 (74.2)	67 (25.8)	ref.	40 (15.4)	ref.	30 (11.5)	ref.
	Yes	14 (66.7)	7 (33.3)	1.27 [0.47, 3.42] ^f^	5 (23.8)	1.40 [0.44, 4.39] ^f^	3 (14.3)	0.90 [0.22, 3.63] ^f^
Poor quality of sleep ^b^								
	No	78 (80.4)	19 (19.6)	ref.	15 (15.5)	ref.	11 (11.3)	ref.
	Yes	129 (70.1)	55 (29.9)	1.72 [0.94, 3.17] ^f^	30 (16.3)	1.16 [0.57, 2.34] ^f^	22 (12.0)	1.09 [0.48, 2.45] ^f^

5-FU, 5-fluorouracil; CI, confidence interval; CIPN, chemotherapy induced peripheral neuropathy; OR, odds ratio. * *p* < 0.05, ** *p* < 0.01, *** *p* < 0.001. ^a^ Baseline depression and anxiety were defined as presenting the respective sub-score equal to or higher than 11 in the Hospital Anxiety and Depression Scale. ^b^ Poor quality of sleep at baseline was defined as presenting a total score equal to or higher than five in the Pittsburg Sleep Quality Index. ^c^ Adjusted for age. ^d^ Adjusted for age and education. ^e^ Adjusted for age, education, cancer stage and breast cancer subtypes. ^f^ Adjusted for age, education and cancer stage.

**Table 4 healthcare-11-03132-t004:** Association of socio-demographic and clinical characteristics of patients without cognitive impairment before treatment, with the variation in the MoCA score between the follow-up and the baseline evaluations.

		MoCA Value after One Year Minus Baseline Value		MoCA Value after Three Years Minus Baseline Value		MoCA Value after Five Years Minus Baseline Value	
		Mean (SD)	Adjusted β Coefficient [95% CI]	Mean (SD)	Adjusted ꞵ Coefficient [95% CI]	Mean (SD)	Adjusted ꞵ Coefficient [95% CI]
All participants		0.6 (2.4)		0.1 (2.8)		0.3 (2.9)	
MoCA score at baseline			−0.20 *** [−0.26, −0.14]		−0.36 *** [−0.46, −0.26]		−0.38 *** [−0.48, −0.28]
Age (years) ^a^							
	<50	0.5 (2.2)	ref.	0.2 (2.7)	ref.	0.5 (2.3)	ref.
	50–64	0.6 (2.5)	−0.29 [−0.80, 0.23] ^e^	0.0 (2.8)	−0.17 [−0.79, 0.44] ^e^	0.4 (2.9)	−0.15 [−0.78, 0.47] ^e^
	≥65	0.6 (2.6)	−0.74 * [−1.42, −0.06] ^e^	−0.0 (3.1)	−0.19 [−0.96, 0.57] ^e^	−0.3 (3.5)	−0.87 * [−1.65, −0.09] ^e^
Education (years) ^a^							
	≤4	0.7 (2.6)	ref.	−0.2 (3.3)	ref.	0.1 (3.4)	ref.
	5–9	0.5 (2.3)	0.91 ** [0.28, 1.53] ^f^	0.2 (2.5)	1.65 *** [0.91, 2.39] ^f^	0.3 (2.7)	1.25 ** [0.50, 2.01] ^f^
	10–12	0.4 (2.4)	1.33 *** [0.56, 2.10] ^f^	0.1 (2.5)	2.06 *** [1.15, 2.97] ^f^	0.4 (2.2)	1.94 *** [1.02, 2.87] ^f^
	>12	0.4 (1.9)	1.73 *** [0.85, 2.61] ^f^	0.6 (2.1)	2.87 *** [1.82, 3.91] ^f^	0.5 (2.1)	2.38 *** [1.32, 3.44] ^f^
Cancer-stage							
	0/I	0.6 (2.6)	ref.	0.1 (2.9)	ref.	0.1 (3.0)	ref.
	II	0.5 (2.2)	−0.02 [−0.52, 0.49] ^g^	0.0 (2.5)	−0.21 [−0.80, 0.38] ^g^	0.5 (2.7)	0.28 [−0.32, 0.88] ^g^
	III	0.6 (2.4)	0.08 [−0.58, 0.73] ^g^	−0.2 (3.2)	−0.44 [−1.21, 0.33] ^g^	0.5 (2.9)	0.29 [−0.49, 1.08] ^g^
Subtypes ^b^							
	ER+/HER2	0.5 (2.5)	ref.	0.0 (2.9)	ref.	0.2 (3.0)	ref.
	HER2+	0.5 (2.1)	0.03 [−0.62, 0.68] ^g^	0.0 (2.6)	−0.02 [−0.79, 0.75] ^g^	0.6 (2.4)	0.44 [−0.34, 1.22] ^g^
	Triple negative	0.2 (2.3)	−0.39 [−1.23, 0.45] ^g^	−0.5 (2.9)	−0.60 [−1.59, 0.39] ^g^	0.0 (2.6)	−0.36 [−1.37, 0.65] ^g^
Chemotherapy							
	No	0.8 (2.6)	ref.	0.2 (2.9)	ref.	0.2 (3.1)	ref.
	Yes	0.4 (2.3)	−0.32 [−0.78, 0.15] ^g^	−0.0 (2.8)	−0.32 [−0.87, 0.23] ^g^	0.3 (2.8)	0.02 [−0.54, 0.58] ^g^
Radiotherapy							
	No	0.9 (2.3)	ref.	0.4 (2.7)	ref.	0.4 (3.1)	ref.
	Yes	0.4 (2.5)	−0.19 [−0.69, 0.31] ^g^	−0.1 (2.9)	−0.28 [−0.87, 0.31] ^g^	0.3 (2.8)	0.01 [−0.59, 0.62] ^g^
Hormone therapy							
	No	0.6 (2.0)	ref.	0.2 (2.7)	ref.	0.3 (2.6)	ref.
	Yes	0.5 (2.5)	0.09 [−0.51, 0.68] ^g^	0.0 (2.9)	−0.02 [−0.73, 0.69] ^g^	0.3 (3.0)	0.17 [−0.55, 0.89] ^g^
Anxiety ^c^							
	No	1.0 (2.4)	ref.	0.5 (2.8)	ref.	0.7 (2.9)	ref.
	Yes	0.3 (2.5)	−0.63 ** [−1.07, −0.19] ^g^	−0.1 (2.9)	−0.48 [−1.01, 0.05] ^g^	0.0 (2.9)	−0.71 ** [−1.25, −0.18] ^g^
Depression ^c^							
	No	0.7 (2.5)	ref.	0.3 (2.8)	ref.	0.5 (2.9)	ref.
	Yes	0.5 (2.5)	−0.47 [−1.28, 0.34] ^g^	−0.4 (3.5)	−1.13 * [−2.08, −0.18] ^g^	−0.5 (3.2)	−1.60 ** [−2.55, −0.64] ^g^
Poor quality of sleep ^d^							
	No	1.0 (2.2)	ref.	0.6 (2.8)	ref.	1.0 (2.6)	ref.
	Yes	0.6 (2.6)	−0.23 [−0.70, 0.23] ^g^	0.1 (2.9)	−0.49 [−1.04, 0.06] ^g^	0.2 (3.1)	−0.68 * [−1.23, −0.12] ^g^

CI, confidence interval; MoCA, Montreal Cognitive Assessment; SD, standard deviation. * *p* < 0.05, ** *p* < 0.01, *** *p* < 0.001. ^a^ Categories of age and education as they are used in the classification for cognitive impairment based on normative data. ^b^ This information is missing for 24 participants. ^c^ Baseline depression and anxiety were defined as presenting the respective sub-score equal to or higher than 11 in the Hospital Anxiety and Depression Scale. ^d^ Poor quality of sleep at baseline was defined as presenting a total score equal to or higher than five in the Pittsburg Sleep Quality Index. ^e^ Adjusted for MoCA score at baseline. ^f^ Adjusted for MoCA score at baseline and for age. ^g^ Adjusted for MoCA score at baseline, age and education.

## Data Availability

The datasets generated and analyzed in this study will not be publicly available, given that the included patients do not specifically provide their consent for public sharing of their data and that anonymization is unlikely to be feasible since the identification of patients treated in only one institution within a relatively short period may be possible when taking sociodemographic and clinical characteristics into account.

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
