# Peer review of "Cancer-Related Neuropathic Pain, Chemotherapy-Induced Peripheral Neuropathy and Cognitive Decline in a 5-Year Prospective Study of Patients with Breast Cancer—NEON-BC"

_healthcare, 2023, doi:10.3390/healthcare11243132_

Round 1

Reviewer 1 Report

Comments and Suggestions for Authors

In general, this is a valuable study providing information which is helpful to clinicians in understanding overall expectations, and to use in advising patients. My concerns about it are more along the lines of missed opportunities, not fatal flaws. I truly appreciated the opportunity to review this study.

Title: Fine

Introduction:  Fine

Materials and Methods: Good. Very systematic regarding time points of evaluation and choice of outcome tools, for domains being assessed including CIPN, neuropathic pain, and cognition. Of note there is only about an 8% dropout rate, which is very strong. It is not stated whether the neurologist(s) conduction the examinations were blinded or not, so I assume not; also the fact that one of the assessment time points was “3 weeks after chemotherapy” also indicates non-blinded status. Overall the methodology is good. One point that needs to be explicitly clarified is regarding the intent of the “neuropathic pain” category, as it is unclear to me whether this category relates to locoregional (“postmastectomy”) neuropathic pain, to generalized CIPN neuropathic pain, and whether any other possible neuropathic pain (such as a contralateral carpal tunnel syndrome or radiculopathy) might be included.

Results:

Table 1. A few clarifications are requested. 1) What level does “education (years)” signify? Is this all education (including primary years), or secondary or postsecondary? If all education, 42% having 4 years of less seems very low. 2) Are there criteria for “practicing physical activity”?

3.1 Neuropathic pain. In lines 218-219 it is stated that pain was localized to the breast, arm and axilla in significant percentages of patients. Did this category mainly consist of locoregional pain issues? What percentages of individuals have neuropathic pain from CIPN or from other (possible) reasons? If this data was not explicitly collected at least have a statement about generally what symptom distributions were included, either here or in the discussion.

3.2 CIPN. This is fine but it would have also been interesting to see polyneuropathy data for the overall group, ie not just those who received chemotherapy. In my experience (including having been involved in a prospective study in which we identified if polyneuropathy was present in our subjects as part of baseline data collection) many cancer patients do have underlying polyneuropathy from health comorbidities.

Figure 2. The baseline MoCA scores are reported but not the baseline neuropathic pain or CIPN scores. Why are these not included in the figure? This is a significant limitation, as patients might have underlying conditions that produce neuropathy or neuropathic pain, or theoretically might even have these problems pretreatment from the cancer. Per Methods, the data was collected so it is surprising not to see it here.

Discussion:

1)      The most fascinating result of this study in my opinion is regards to the arc of neuropathic pain reporting, getting worse before it gets better, ie peaking at the 3 year time point rather than progressively improving between years 1,3, and 5 as might be expected (and is indeed seen with the CIPN data). This pattern rings true to me but it is hard to say why this is happening. Do the authors have any observations about their population that might explain the neuropathic pain reporting peaking at year 3?

2)      While there is some discussion of clinical risk factors for the various outcomes (NP, CIPN, cognition), most valuable would be regarding the clinical characteristics of patients who continue to experience these difficulties at year 5. This topic is not explicitly or cohesively discussed and would be a very helpful addition in terms of guiding clinicians in being able to identify patients at highest risk of having these problems long term. (The abstract does mention some associations such as cancer stage, triple negative status and taxanes with the various outcomes but this seems to relate to all data not just the long term subset.)

Author Response

Point-by-point response to the Reviewers’ comments

Reviewer #1 comment #1:

In general, this is a valuable study providing information which is helpful to clinicians in understanding overall expectations, and to use in advising patients. My concerns about it are more along the lines of missed opportunities, not fatal flaws. I truly appreciated the opportunity to review this study.

Reply to Reviewer #1 comment #1:

We thank the Reviewer for the favorable comment.

Reviewer #1 comments #2 and #3:

Title: Fine

Introduction:  Fine

Reply to Reviewer #1 comments #2 and #3:

No specific comment or suggestion were made.

Reviewer #1 comment #4:

Materials and Methods: Good. Very systematic regarding time points of evaluation and choice of outcome tools, for domains being assessed including CIPN, neuropathic pain, and cognition. Of note there is only about an 8% dropout rate, which is very strong.

Reply to Reviewer #1 comment #4:

No specific comment or suggestion were made.

Reviewer #1 comment #5:

It is not stated whether the neurologist(s) conduction the examinations were blinded or not, so I assume not; also the fact that one of the assessment time points was “3 weeks after chemotherapy” also indicates non-blinded status.

Reply to Reviewer #1 comment #5:

The neurologist was not blinded regarding treatments and clinical characteristics of the patient at the moment of the neurological evaluation. Unfortunately, the team of neurologists was too small to allow a separation between the systematic clinical evaluation within the study and the provision of clinical care to the participant/patient, but standardized procedures were followed to minimize bias.

This is now being acknowledged in the discussion of the revised manuscript (please see lines 536-541).

Reviewer #1 comment #6:

Overall the methodology is good. One point that needs to be explicitly clarified is regarding the intent of the “neuropathic pain” category, as it is unclear to me whether this category relates to locoregional (“postmastectomy”) neuropathic pain, to generalized CIPN neuropathic pain, and whether any other possible neuropathic pain (such as a contralateral carpal tunnel syndrome or radiculopathy) might be included.

Reply to Reviewer #1 comment #6:

The revised version includes a more clarifying statement in lines 179-183: “In the present study, only CRNP was analyzed. Therefore, only neuropathic pain which onset was after breast cancer treatment and in locations related to the breast and axillary surgeries and to radiotherapy-treated areas was considered, as well as locations related to breast reconstruction surgery and neuropathic pain due to CIPN (one women with pain in the feet and another one in tips of fingers and toes).”

Reviewer #1 comment #7:

Results:

Table 1. A few clarifications are requested. 1) What level does “education (years)” signify? Is this all education (including primary years), or secondary or postsecondary? If all education, 42% having 4 years of less seems very low. 2) Are there criteria for “practicing physical activity”?

Reply to Reviewer #1 comment #7:

In Table 1, three categories of education are reported: four years or less of schooling years (primary education or level 1 of the International Standard Classification of Education [ISCE]), between five and nine years of schooling years (ISCE level 2), and at least ten years (at least ISCE level 3). Indeed, 42% of participants with four or less than four years of schooling is a low level of education for the cohort but it is similar to the general Portuguese population, when considering the age distribution of the cohort. The Portuguese education system has increased the minimum years of mandatory schooling attendance over the past 50 years but for many participants of the cohort, only four years of schooling were mandatory at the time they were attending school.

We have updated Table 1 with the indication of education levels.

The practice of physical activity before breast cancer diagnosis was only assessed at the three-year evaluation. Indeed, as the first year of the study included many evaluations of the participants, namely at baseline, post-surgery, post-chemotherapy, six months after the diagnosis of cancer-related neuropathic pain and CIPN, and one-year after the baseline assessment, the study focused in the first year on the robust identification of cancer-related neuropathic pain and CIPN and less on lifestyle characteristics to spare women an excessively long assessment in a period already filled with many consultations, treatments and exams at the hospital. The study could continue after the first year of follow-up, and participants were asked at the three-year evaluation regarding current time and before breast cancer diagnosis if they practiced any sport, what sport, frequency and time spent at this activity. For this analysis we dichotomized the practice of physical activity before breast cancer diagnosis as “playing a sport, yes or no”. As the WHO refers physical activity as “all movement including during leisure time, for transport to get to and from places, or as part of a person’s work.”, our classification does not cover transportation and work activity. Therefore, we changed “physical activity” to “playing a sport”. We added a more detailed description of data collection in lines 120-131.

Reviewer #1 comment #8:

3.1 Neuropathic pain. In lines 218-219 it is stated that pain was localized to the breast, arm and axilla in significant percentages of patients. Did this category mainly consist of locoregional pain issues? What percentages of individuals have neuropathic pain from CIPN or from other (possible) reasons? If this data was not explicitly collected at least have a statement about generally what symptom distributions were included, either here or in the discussion.

Reply to Reviewer #1 comment #8:

At the one- and three-year evaluations, there were one woman with neuropathic pain in her feet and another one in her fingers and toes due to CIPN but there were no cases of neuropathic pain due to CIPN at the five-year assessment. Nevertheless, we modified the sentence and added more information on the localization of the neuropathic pain related with the oncologic disease, as follows: “At the five-year evaluation, CRNP was localized in the breast (68% of women with CRNP), the arm (21.3%), the axillary area (16.0%), the chest wall (6.7%), and the inguinal region (2.7%).“ (lines 268-270)

The neurological exam performed at all evaluations, including before cancer treatments and after five years, identified several neurological problems manifesting or not with pain, such as, and naming only the most frequent, tension headaches, migraines, tremor, and carpal tunnel syndrome. However, this study focuses on the most frequent breast cancer-related neurological complications: CRNP, CIPN, and cognitive impairment.

Reviewer #1 comment #9:

3.2 CIPN. This is fine but it would have also been interesting to see polyneuropathy data for the overall group, ie not just those who received chemotherapy. In my experience (including having been involved in a prospective study in which we identified if polyneuropathy was present in our subjects as part of baseline data collection) many cancer patients do have underlying polyneuropathy from health comorbidities.

Reply to Reviewer #1 comment #9:

All participants were evaluated with a neurological exam before any treatment to identify pre-existing neurological problems, namely neuropathic pain, polyneuropathy, and cognitive impairment. CIPN was then defined as “… as peripheral neuropathy diagnosed after chemotherapy or worsening of a preexisting neuropathy after chemotherapy. “(lines 164-165).

We added Table S2 in the supplementary material presenting the different neurological conditions identified at baseline.

Reviewer #1 comment #10:

Figure 2. The baseline MoCA scores are reported but not the baseline neuropathic pain or CIPN scores. Why are these not included in the figure? This is a significant limitation, as patients might have underlying conditions that produce neuropathy or neuropathic pain, or theoretically might even have these problems pretreatment from the cancer. Per Methods, the data was collected so it is surprising not to see it here.

Reply to Reviewer #1 comment #10:

We are now adding TableS2 which presents the neurological conditions identified during the baseline neurological exam, and we refer in lines 245-248 the most relevant data regarding neuropathy and neuropathic pain, namely the presence of diabetic polyneuropathy in three women, one participant with tomaculous neuropathy, and three patients with carpal tunnel syndrome.

Nevertheless, we consider that these conditions should not be represented in Figure 2 as they were considered as not cancer-related during the neurological exam.

Figure 2 was updated: “neuropathic pain” was changed to “cancer-related neuropathic pain” and the baseline bar with 0% was added for CRNP and CIPN.

We updated neuropathic pain (NP) to cancer-related neuropathic pain (CRNP) throughout the manuscript.

Reviewer #1 comment #11:

Discussion:

  • The most fascinating result of this study in my opinion is regards to the arc of neuropathic pain reporting, getting worse before it gets better, ie peaking at the 3 year time point rather than progressively improving between years 1,3, and 5 as might be expected (and is indeed seen with the CIPN data). This pattern rings true to me but it is hard to say why this is happening. Do the authors have any observations about their population that might explain the neuropathic pain reporting peaking at year 3?

Reply to Reviewer #1 comment #11:

The prevalence of CRNP at three years –  24.0% (95%CI: 20.2%,28.2%) – is not statistically different from the estimate 21.0% (95%CI: 17.4%,25.0%) at the one-year evaluation, as the 95% overlapping confidence intervals indicate. The prevalence of CRNP did decrease five years after breast cancer diagnosis.

The severity of CRNP increased from the one- to the three-year evaluation and then there were no statistical differences between the three- and the five-year evaluations. No improvement was observed for CRNP over time. We added this important interpretation of the results in the discussion, in lines 372-376.

Reviewer #1 comment #12:

  • While there is some discussion of clinical risk factors for the various outcomes (NP, CIPN, cognition), most valuable would be regarding the clinical characteristics of patients who continue to experience these difficulties at year 5. This topic is not explicitly or cohesively discussed and would be a very helpful addition in terms of guiding clinicians in being able to identify patients at highest risk of having these problems long term. (The abstract does mention some associations such as cancer stage, triple negative status and taxanes with the various outcomes but this seems to relate to all data not just the long term subset.)

Reply to Reviewer #1 comment #12:

Although the results on risk factors for CRNP, CIPN and cognitive decline at five years are presented in tables 2, 3, and 4, we had not discussed these specific results and we are now adding this discussion in lines 547-567.

Reviewer 2 Report

Comments and Suggestions for Authors

The objective of the study was to quantify the prevalence of Neuropathic pain, chemotherapy-induced peripheral neuropathy, and cognitive impairment up to five years after diagnosis of breast cancer, as well as by assessing their determinants.

The study has a strong descriptive characteristic. This may not be a problem if your justification is solid. That's not the case. The introduction does not present elements to attest to its innovation. In this sense, it seems that the study emerges from the reuse of already published data, modified only by the inclusion of two more years in the analysis, following the sentences presented in lines 57-59.

Not enough, its justification is based on a systematic review published in 2014 (reference 6). Neuropathic pain, chemotherapy-induced peripheral neuropathy, and cognitive impairment are parameters regularly studied in patients with breast cancer. Therefore, the justification for the present study is weak, and new elements that could impact this context were not presented.

As for methods, the study monitors few parameters during follow-up. Several systematic reviews and meta-analyses demonstrate beneficial effects of physical exercise (for example) in chemotherapy-induced peripheral neuropathy. Both regular physical exercise and dietary recalls, sleep quality analyzes (and so on...) should have been carried out. Otherwise, it is difficult to associate it directly and solely with the disease.

Author Response

Point-by-point response to the Reviewer’s comments

REVIEWER #2

Reviewer #2 comment #1:

The objective of the study was to quantify the prevalence of Neuropathic pain, chemotherapy-induced peripheral neuropathy, and cognitive impairment up to five years after diagnosis of breast cancer, as well as by assessing their determinants.

The study has a strong descriptive characteristic. This may not be a problem if your justification is solid. That's not the case. The introduction does not present elements to attest to its innovation. In this sense, it seems that the study emerges from the reuse of already published data, modified only by the inclusion of two more years in the analysis, following the sentences presented in lines 57-59.

Reply to Reviewer #2 comment #1:

In the introduction, lines 52-64, we are now updating some references and highlighting the gaps in the literature that demonstrate the unique contribution of this study in terms of its robust methodology. Additionally, in lines 73-79, we added the specific interest in updating the previous results with data on five years after breast cancer diagnosis, an important moment in the clinical follow-up of patients, as part of them will be discharged from the cancer hospital and will be followed by their primary care physicians. Therefore, we consider this knowledge important for oncology teams and healthcare professionals of primary care units to be aware of the frequency of these neurological complications of breast cancer.

Reviewer #2 comment #1:

Not enough, its justification is based on a systematic review published in 2014 (reference 6). Neuropathic pain, chemotherapy-induced peripheral neuropathy, and cognitive impairment are parameters regularly studied in patients with breast cancer. Therefore, the justification for the present study is weak, and new elements that could impact this context were not presented.

Reply to Reviewer #2 comment #2:

We updated the references and corresponding scientific evidence in lines 52-64.

Reviewer #2 comment #3:

As for methods, the study monitors few parameters during follow-up. Several systematic reviews and meta-analyses demonstrate beneficial effects of physical exercise (for example) in chemotherapy-induced peripheral neuropathy. Both regular physical exercise and dietary recalls, sleep quality analyzes (and so on...) should have been carried out. Otherwise, it is difficult to associate it directly and solely with the disease.

Reply to Reviewer #2 comment #4:

We are now adding Tables S6, S7, and S8 in the supplementary material, comparing the frequency of socio-demographic, lifestyles, and co-morbidities in the different groups of participants regarding the presence of CIPN over time. Only fruits and vegetables consumption of at least five portions per day was significantly associated with a lower frequency of CIPN present in all evaluations.

We report this result in lines 341-343 and Table 3, and discuss it in lines 391-403.

We also present data collection of physical activity (lines 127-131) and discuss the limitations of the present study regarding the association between this exposure and CIPN in lines 526-530.

We had already presented the results on sleep quality and CIPN in Table 3.

Round 2

Reviewer 2 Report

Comments and Suggestions for Authors

My starting position is maintained.

Author Response

We thanks the reviewer for the review of our manuscript and we provide a point-by-point response as follows:

Reviewer #2 comment #1:

The objective of the study was to quantify the prevalence of Neuropathic pain, chemotherapy-induced peripheral neuropathy, and cognitive impairment up to five years after diagnosis of breast cancer, as well as by assessing their determinants.

The study has a strong descriptive characteristic. This may not be a problem if your justification is solid. That's not the case. The introduction does not present elements to attest to its innovation. In this sense, it seems that the study emerges from the reuse of already published data, modified only by the inclusion of two more years in the analysis, following the sentences presented in lines 57-59.

Reply to Reviewer #2 comment #1:

In the introduction, lines 52-64, we are now updating some references and highlighting the gaps in the literature that demonstrate the unique contribution of this study in terms of its robust methodology. Additionally, in lines 73-79, we added the specific interest in updating the previous results with data on five years after breast cancer diagnosis, an important moment in the clinical follow-up of patients, as part of them will be discharged from the cancer hospital and will be followed by their primary care physicians. Therefore, we consider this knowledge important for oncology teams and healthcare professionals of primary care units to be aware of the frequency of these neurological complications of breast cancer.

Reviewer #2 comment #1:

Not enough, its justification is based on a systematic review published in 2014 (reference 6). Neuropathic pain, chemotherapy-induced peripheral neuropathy, and cognitive impairment are parameters regularly studied in patients with breast cancer. Therefore, the justification for the present study is weak, and new elements that could impact this context were not presented.

Reply to Reviewer #2 comment #2:

We updated the references and corresponding scientific evidence in lines 52-64.

Reviewer #2 comment #3:

As for methods, the study monitors few parameters during follow-up. Several systematic reviews and meta-analyses demonstrate beneficial effects of physical exercise (for example) in chemotherapy-induced peripheral neuropathy. Both regular physical exercise and dietary recalls, sleep quality analyzes (and so on...) should have been carried out. Otherwise, it is difficult to associate it directly and solely with the disease.

Reply to Reviewer #2 comment #4:

We are now adding Tables S6, S7, and S8 in the supplementary material, comparing the frequency of socio-demographic, lifestyles, and co-morbidities in the different groups of participants regarding the presence of CIPN over time. Only fruits and vegetables consumption of at least five portions per day was significantly associated with a lower frequency of CIPN present in all evaluations.

We report this result in lines 341-343 and Table 3, and discuss it in lines 391-403.

We also present data collection of physical activity (lines 127-131) and discuss the limitations of the present study regarding the association between this exposure and CIPN in lines 526-530.

We had already presented the results on sleep quality and CIPN in Table 3.
